# Phylogeographic and Paleoclimatic Modelling Tools Improve Our Understanding of the Biogeographic History of *Hierophis viridiflavus* (Colubridae)

**DOI:** 10.3390/ani13132143

**Published:** 2023-06-29

**Authors:** Iñaki Romero-Iraola, Inês Freitas, Yolanda Jiménez-Ruíz, Philippe Geniez, Mario García-París, Fernando Martínez-Freiría

**Affiliations:** 1Departamento de Herpetología, Sociedad de Ciencias Aranzadi, Paseo de Zorroaga 11, 20004 Donostia-San Sebastián, Spain; 2Museo Nacional de Ciencias Naturales, Centro Superior de Investigación Cinetífica (CSIC), José Gutiérrez Abascal 2, 28006 Madrid, Spain; 3Centro de Investigação em Biodiversidade e Recursos Genéticos (CIBIO), InBIO Laboratório Associado, Campus de Vairão, Universidade do Porto, 4485-661 Porto, Portugal; 4BIOPOLIS Program in Genomics, Biodiversity and Land Planning, Centro de Investigação em Biodiversidade e Recursos Genéticos (CIBIO), Campus de Vairão, 4485-661 Porto, Portugal; 5Centre d´Ecologie Fonctionnelle & Evolutive (CEFE), Ecole Pratique des Hautes Etudes (EPHE-PSL), Centre National de la Recherche Scientifique (CNRS), Biogéographie et Ecologie des Vertébrés, Université de Montpellier, 34293 Montpellier, France

**Keywords:** reptiles, European whip snake, ecological niche models, glacial refugia, genetic diversity

## Abstract

**Simple Summary:**

Previous paleoclimatic modelling studies on the European whip snake, *Hierophis viridiflavus*, hardly explain its genetic structure. Through the combination of phylogeographic and paleoclimatic modelling analyses, we reconstructed the biogeographic history of the species. Phylogeographic analyses recovered two major lineages that had diversified in different ways in the late Pleistocene. Paleoclimatic models unveiled the species range dynamics since the late Pleistocene, showing major range contractions into multiple climatic refugia in Southern Europe during glacial and interglacial events, as well as more recent northwards expansions. This study contributes to our knowledge of *H. viridiflavus’* historical biogeography and sheds more light on the evolutionary processes that took place in the Mediterranean Basin hotspot.

**Abstract:**

Phylogeographic and paleoclimatic modelling studies have been combined to infer the role of Pleistocene climatic oscillations as drivers of the genetic structure and distribution of Mediterranean taxa. For the European whip snake, *Hierophis viridiflavus*, previous studies based on paleoclimatic modelling have depicted a low reliability in the pattern of past climatic suitability across the central Mediterranean Basin, which barely fits the species’ genetic structure. In this study, we combined phylogeographic and paleoclimatic modelling tools to improve our understanding of the biogeographic history of *H. viridiflavus*, particularly extending the sampling and phylogeographic inferences to previously under-sampled regions. Phylogeographic analyses recovered two major clades that diverged at the beginning of the Pleistocene and had diversified in different ways by the late Pleistocene: the east clade (composed of three subclades) and the west clade (with no further structure). Paleoclimatic models highlighted the temperate character of *H. viridiflavus*, indicating range contractions during both the last inter-glacial and last glacial maximum periods. Range expansions from southern-located climatic refugia likely occurred in the Bølling–Allerød and Middle Holocene periods, which are supported by signals of demographic growth in the west clade and South–East–North subclade. Overall, this work improves our understanding of the historical biogeography of *H. viridiflavus*, providing further insights into the evolutionary processes that occurred in the Mediterranean Basin hotspot.

## 1. Introduction

Climatic variation is one of the major drivers determining species distribution patterns [1]; it shapes the expression of ecological niches across species ranges [2] and acts over time, partially determining the current patterns of genetic structure and distribution of distinct taxa worldwide [3,4,5]. Many phylogeographic studies have recurrently linked the genetic signatures of distinct taxa to the repeated warming–cooling cycles that occurred during the Pleistocene, e.g., [6,7,8]. In different ways, these cycles promoted species range expansions or retractions (in accordance with species ecological requirements) as well as population isolation in areas of suitable climatic conditions (i.e., climatic refugia) [9]. Consequently, they lead to important demographic processes such as population bottlenecks and expansions and thus affect the genetic diversity of species [3,4].

The Mediterranean Basin is a compelling area for investigating biogeographical patterns due to its high levels of biodiversity and endemicity due to a turbulent paleo-tectonic and climatic history and the occurrence of marked environmental gradients [4,10,11]. During the last years, research aimed at explaining the factors involved in biogeographic patterns in the region has been notorious, which has in part been mediated by the increasingly frequent combination of genetic and ecological tools and the use of ectothermic, low dispersal taxa as model systems [12,13]. Remarkably, our understanding of the evolutionary processes that occurred in the Pleistocene has been greatly improved through the application of ecological models to reconstruct past range dynamics (i.e., paleoclimatic modelling) [14]. Research relying on paleoclimatic modelling has allowed, for instance, for us to link species responses to climatic oscillations and patterns of genetic structure and diversity [15,16], confirming that particular regions such as the Southern Mediterranean peninsulas and the Maghreb acted as major climatic refugia and centers of diversification, e.g., [17,18], or as unveiling corridors for population connectivity, e.g., [19,20].

Paleoclimatic models are correlative approaches that establish links between species occurrences and current climatic conditions, and they make predictions into past scenarios [14,21]. Consequently, they are sensitive to important issues such as the similitude between the environmental conditions of current and past scenarios [22], the assumption that the species is in equilibrium with the environment [23,24] or the representativeness of occurrence data in relation to the species range [25,26]. Beyond these considerations, paleoclimatic models are particularly meaningful when both the genetic signal (i.e., divergence times) and the paleoclimatic scenarios present a temporal match or the latter at least represents the major climatic events for a particular period [15,18].

The European whip snake, *Hierophis viridiflavus* (Lacépède, 1789), is a medium-sized thermophilic colubrid snake with a south-central European para-Mediterranean distribution, occurring across a variety of open environments that are characterized by temperate climatic conditions [27,28,29,30]. Phylogeographic studies have pointed that the species diverged from its phylogenetically sister species *Hierophis gemonensis* (Laurenti, 1768) in the late Miocene (about 7 Mya) and that posteriorly, in the Pleistocene (about 2.2 Mya) [31], it diversified into two major clades: (1) a western clade, which distributed from west-central Italy, including Corsica and Sardinia to the Spanish Pyrenees; and (2) an eastern clade, occurring from Southern Italy, including Sicily up to the Alps and along the Adriatic coast up to Dalmatia. These clades were sometimes considered as distinct species, *H. viridiflavus* and *H. carbonarius* (Bonaparte, 1833) [31,32]; however, a recent review on the taxonomy of European herpetofauna assigned the subspecies rank for these two clades based on incomplete isolation between them [33].

Recent research to infer scenarios for the evolution of the Mediterranean whip snakes (genus *Hierophis*) using paleoclimatic models has indicated a major response of these taxa to Pleistocene glacial events [32]. However, predictions resulting from these models have depicted a low reliability in terms of the pattern of current and past suitability for the three studied taxa, where extensive areas acting as climatic refugia across the Southern European peninsula and North Africa barely fit the genetic inferences [31,32]. Three major issues can be identified as potential factors affecting the reliability of paleoclimatic inferences: (1) the consideration of *H. viridiflavus* and *H. carbonarius* as full species instead of intraspecific lineages, resulting in their use as independent units in the modelling approach, which neglects the assumption that the species is in equilibrium with the environment [23]; (2) the use of a reduced number of occurrence records to perform the models, which are not representative of the species range and ubiquity [25,26]; or (3) the use of paleoclimatic scenarios that are limited to the last glacial maximum, leading to low representativeness of the climatic events that occurred in the Pleistocene and thus limiting our understanding of the evolutionary processes that occurred within the species. 

In this study, we combined phylogeographic and paleoclimatic tools to improve our understanding of the biogeographic history of *H. viridiflavus*, reconstructing a plausible scenario for the evolution of this species since the Pleistocene. Specifically, we aimed to: (i) improve the previous characterization of the intraspecific genetic structure and variability of *H. viridiflavus*, particularly by extending sampling and phylogeographic inferences to previously under-sampled regions; (ii) infer the species range dynamics since the last cycle of the Pleistocene to current times; and (iii) identify the stable climatic areas that likely acted as climatic refugia for the species over time. Considering the thermophilic character of *H. viridiflavus* [27], its occurrence across temperate environments [28,30] and its current genetic structure [31,32], we expect that the species underwent major range contraction into multiple climatic refugia in Southern Europe during glacial events. On the other hand, a likely recent northwards expansion with the increase in temperature was expected to be recovered with the genetic data as well.

## 2. Material and Methods

### 2.1. Sampling

Previous studies mostly focused on the Italian populations of *H. viridiflavus*; thus, many areas across North-eastern Spain, France, and Corsica were under-sampled [31,32,34,35,36]. To improve previous phylogeographic inferences, tissue samples from populations across these aforementioned areas were collected, which represented a total of 80 individuals of *H. viridiflavus* (Figure 1).

Samples were obtained from: (i) live specimens captured along transects and fieldwork campaigns across Euskadi (*n* = 14) [38] and Aragon (*n* = 4), and (ii) ethanol-preserved specimens from four museum collections (*n* = 62) (for details, see Appendix A). All samples were preserved in ethanol.

A total of 4158 occurrence records of *H. viridiflavus* (at a 5 arc-minutes resolution, ∼10 × 10 km resolution) were used to model the ecological niche of the species (Figure 1 and Appendix A). Occurrence records were obtained from the online GBIF Portal (*n* = 3853; https://www.gbif.org/es/species/2458916, accessed on 25 June 2023) and from personal (*n* = 304) datasets. The GBIF dataset was carefully examined to delete records that were potentially erroneous, which were determined as those falling outside the species range determined by the IUCN [37].

### 2.2. Climatic Variables

Nineteen bioclimatic variables for the current conditions at a 5 arc-minutes resolution (∼10 × 10 km) were downloaded from the Paleoclim portal (http://www.paleoclim.org/, accessed on 25 June 2023), clipped to a study area consisting of the species range plus a 200 km buffer, and considered as the initial set of predictors in our ecological modelling approach. Two of these variables (BIO18 and BIO19) were discarded due to the presence of spatial artefacts located in the Iberian peninsula, e.g., [19,39], while the remaining seventeen bioclimatic variables were tested for correlations [40]. Only four low-correlated variables (R < 0.6; VIF < 1.7) were retained for further ecological analyses: BIO3 (isothermality), BIO5 (maximum temperature of warmest month), BIO12 (annual precipitation), and BIO15 (precipitation seasonality). Some of these variables have been previously used to model the distribution of *H. viridiflavus* [31] and other species of snakes, e.g., [15,39,41]. Therefore, they are expected to act as informative predictors for inferring the species’ ecological niche and potential distribution.

The past conditions used to project the models were downloaded from the Paleoclim portal (http://www.paleoclim.org/, accessed on 25 June 2023) at a 5 arc-minutes resolution (∼10 × 10 km) and included three periods in the Pleistocene, i.e., the last interglacial (LIG; ~120,000–140,000 years BP), last glacial maximum (LGM; ~21,000 years BP) and Bølling–Allerød period (BA; ~14.7–12.9 years BP), as well as one period in the Holocene, i.e., the mid-Holocene (MH; ~8.326–4.2 years BP). The LIG and LGM are widely recognized as being qualitatively representative of previous cycles during the Pleistocene [42], and together with the MH, have been successfully used in previous studies to investigate the historical biogeography of other snake species [15,19,39]. In this work, we extended model projections to an additional period of the Pleistocene (i.e., the Bølling–Allerød period, BA), which enables us to assess the general warming trend that occurred during that period until the Early Holocene [43] that may have affected this species’ historical demography and population structure.

### 2.3. Phylogeographic Analyses

A total of 224 *H. viridiflavus* ND4 (NADH dehydrogenase subunit 4) sequences were considered to estimate the phylogenetic relationships and create haplotype networks (Appendix A). We generated 80 unpublished sequences in this work (GenBank accession numbers to be added after acceptance) (see Appendix A for laboratory procedures), and 144 sequences were retrieved from previous studies (GenBank accession numbers: LN552045-LN552095, [31]; FJ430621-FJ430660, [34]; KY923281-KY923287, [35]; and MW297553–MW297679, [36]; see Appendix A). *Hierophis gemonensis* (Laurenti, 1768), considered to be the closest related species to *H. viridiflavus* [31], was included as an outgroup (accession number: LN552113, [31]) in the phylogenetic analyses. Sequences were manually aligned and edited using Geneious v 4.8.5 [44]. A complete alignment of 834 bp was used for phylogenetic inferences and a shorter fragment of 568 bp with no missing data was used to generate haplotype networks.

Phylogenetic relationships were inferred using a Bayesian inference (BI) method implemented in BEAST v 1.7.5 [45]. The best-fit partitioning scheme and substitution model were selected using the Bayesian information criterion (BIC) in PartitionFinder v1.1 [46]. The best inferred model, TrN+I without partition into codon positions, and a strict molecular clock were implemented. A coalescence constant population size model had been previously used as a tree model. Considering the absence of reliable calibration dates and fossil records for *Hierophis*, the time of divergence between the main lineages was estimated using two secondary calibrations obtained from [31]: (1) the split of *H. gemonensis* and *H. viridiflavus*, dated at 7 Mya; and (2) the divergence between *H. v. viridiflavus* and *H. v. carbonarius*, estimated at 2.2 Mya. We used a normal prior with a mean that was equal to the node age and a standard deviation adjusted to a 95% confidence interval including ±10% of the mean value to account for uncertainty in age estimates. Three independent runs of 100 million generations were performed, where the sampling trees and parameters estimate every 10,000 generations with 10% of the trees being discarded as burn-in. Convergence of the chains was confirmed in Tracer v.1.7.1 [47]. Trees obtained from multiple independent runs were combined using LogCombiner v 1.7.5. [45], and a summary tree was generated using TreeAnnotator v1.7.1 and visualized in FigTree v 1.4.1 [45].

To assess the genetic structure and diversity within *H. viridiflavus*, the TCS network approach as implemented in PopART v1.7 [48] was used to build the haplotype networks.

To detect past demographic events such as population expansions, Tajima’s D and Fu and Li’s D neutrality tests were performed for the western (*H. v. viridiflavus*) and eastern (*H. v. carbonarius*) clades and for the three eastern lineages, as reflected in our phylogeny. The number of haplotypes, nucleotide diversity (π), and haplotype diversity (Hd) were also calculated for the same groups using DnaSP 5.1 [49].

### 2.4. Paleoclimatic Modelling

To predict the potential species distribution in the current and past conditions, we used the maximum entropy approach, which was implemented in the Maxent ver. 3.3.3k software [50,51]. This algorithm only requires presence data, performs well compared with other methods [52], and has been used successfully in modelling snake species distributions, e.g., [15,19,31].

Occurrence data and climatic variables were imported into Maxent software, where 50 replicates were run with random seeding, 70%/30% training/testing partition, and bootstrap with replacement. Before running the final models, a test calibration process using the Maxent parameters was carried out to address the model parameterization, which was performed using the R package ENMeval [53] (further details in Appendix A). The predictive performance of each model was assessed by measuring the area under the curve of the receiver operating characteristic plots (AUC). The mean percentage contribution to the model of each climatic variable was considered to assess the variable’s importance in explaining the species’ distribution. Univariate response curve profiles for the most important variables were examined to understand the relation between species occurrence and specific predictors, e.g., [15,54].

Standard deviation was used as an indication of prediction uncertainty, e.g., [15,54]. The effect of model extrapolation on predictor variables lying outside of the training range was assessed using a multivariate environmental similarity surface (MESS) map [55]. The MESS values detail the closeness of the site to the distribution of reference points and maps these values across the whole prediction region [55].

Stable climatic areas (i.e., potential areas that could serve as climatic refugia for the species over time) [56] were calculated using the fuzzy overlay approach with the “AND” function on ArcMap 10.6 for every two consecutive time periods [16,57].

## 3. Results

### 3.1. Phylogeographic Analyses

The phylogenetic tree recovered *H. viridiflavus* as a monophyletic group that was composed of two statistically well-supported clades (pp > 0.95; Figure 2 and Appendix A): (1) the west clade (*H. v. viridiflavus*), distributed from North–central–west Italy to France and Spain, also including Corsica and Sardinia; and (2) the east clade (*H. v. carbonarius*), distributed in the rest of the Italian peninsula, Sicily, Slovenia, and Croatia. The west clade has no further statistically supported structure, whereas the east clade consists of three subclades: (1) Sicily, which is distributed through Sicily and the South-west Italian peninsula; (2) South–East–North, which is distributed through the South, East, and North Italian peninsula and North–West Croatia; and (3) Central, which occurs in the Central–West Italian peninsula. The South and Central subclades have high statistical support, whereas the Sicily clade has low statistical support (pp = 0.76; Appendix A). Within-clades diversification occurred at distinct times, first starting with the diversification within the east clade (TMRCA, i.e., time to most recent common ancestor, Avg. = 0.65 Mya, 95% HPD = 0.315–1.022) and later the diversification within the west clade (TMRCA Avg. = 0.27 Mya, 95% HPD = 0.109–0.45) (Figure 2 and Appendix A).

The haplotype network recovered four main haplogroups (Figure 2): one corresponding to the west clade and three other haplogroups corresponding to subclades identified within the east clade. We identified 32 haplotypes, 13 of which came from the West haplogroup, 7 from the Sicily haplogroup, 2 from the Central haplogroup, and 10 from the South–East–North haplogroup (Appendix A). Regarding the west haplogroup, one haplotype (H1) showed a high frequency (93% of the samples in the west clade) and was widely distributed across the whole range (Figure 2). Similarly, within the Central subclade (*n* = 16), 94% of the samples belonged to the H21 haplotype; within the South–East–North subclade (*n* = 76), 72% of the samples belonged to the H32 haplotype, and within the Sicily subclade (*n* = 25), 52% of the samples belonged to the H14 haplotype.

Neutrality tests for the east clade and Sicily and Central subclades were non-significant, pointing to no recent demographic changes. Fu and Li’s D test was significant for the South–East–North subclade with a negative D value of −2.97, indicating a recent population expansion. Similarly, both neutrality tests were significant for the west clade (*H. v. viridiflavus*) with D values of −2.31 and −5.77 (Tajima’s D and Fu and Li’ D, respectively), which points to recent population growth as well (Table 1).

Genetic diversity was higher in the east clade than in the west (Table 1). Thus, the areas holding the highest level of ND4 haplotype diversity are located in Italy, whereas areas with moderate and low diversity were identified to be in Central–Southern France and Northern Spain.

### 3.2. Paleoclimatic Modeling

The model performance was good for both training and testing (Avg. AUC = 0.719, SD = 0.004; Avg. AUC = 0.718, SD = 0.006). One precipitation variable, annual precipitation (BIO12), and one temperature variable, max temperature of warmest month (BIO5), were the variables that most contributed to the distribution of the species (Table 2). The examination of the response curve profiles for these variables indicates that *H. viridiflavus* occurs in temperate areas with moderate to high levels of precipitation (Appendix A).

The model predictions for the current conditions mostly fit the species distribution and also identify suitable areas outside of the current species range (Figure 3 and Appendix A). The standard deviation plots show very low variability between model replicates, meaning that all replicates had similar spatial predictions (Appendix A).

Projections to past scenarios show high variability in the location and extension of areas of climatic suitability (Figure 3 and Appendix A).

The standard deviation plots of the model projections show high values all over the study area. The MESS analyses for each of the projected scenarios show that areas with negative dissimilarity values are reduced (except in the LGM period) and, importantly, that they correspond to areas with low suitability in the average projections (Appendix A).

In the LIG, areas of high climatic suitability were widely located, almost in a similar way to the current times, with the exception of a very restricted suitability across the Italian peninsula and Southern France (Figure 4).

Throughout the LGM, areas of high suitability became restricted to the northeastern part of Sicily, the western coast of Italy, the emerged land between Northeast Italy and Northwest Croatia, the south–central coast of France, and the emerged land in Eastern France. Posterior periods (Bølling–Allerød and mid-Holocene) exhibiting a general trend for an extent of high climatic suitable areas can be observed. Croatia, Italy, Central–Southern France, and Northern Spain were the areas with higher climatic suitability, while Central Europe suffered a reduction in these areas in contrast with the LIG scenario.

The analysis of climatic stability in the suitable conditions while considering consecutive periods indicate Northern Sicily, the Southwestern and Northwestern Italic peninsula, and Southern France as potential climatic refugia for the species in the LIG–LGM (Figure 4). A similar pattern, but with larger areas of stability, was recovered during the LGM–Bølling–Allerød, while a general pattern of large areas of high stability across most of the species current range was recovered after these periods (Figure 4).

## 4. Discussion

In this study, we relied on the combination of phylogeographic and paleoclimatic modelling tools to infer the biogeographic history of *H. viridiflavus*. Previous studies already used these tools for similar purposes; see Refs. [31,32]. However, by increasing the genetic sampling coverage in the western range of the species, performing additional demographic tests on the genetic data, and refining the ecological modelling approach, we provide more detailed inferences about the processes that have shaped the species genetic structure and variability since the Pleistocene.

### 4.1. Genetic Structure and Demography

As previous phylogeographic studies already reported [31,34], our Bayesian phylogeographic tree shows that, after divergence with its sister taxa (*H. gemonensis*), *H. viridiflavus* became structured into two major clades: one distributed from Central Italy to Northeastern Spain (west clade, *H. v. viridiflavus*) and another mostly restricted to Italy (east clade, *H. v. carbonarius*). The west clade has no further supported structure, whereas three subclades were recovered within the east clade (Figure 2 and Appendix A). Overall, the southern half of the Italian peninsula holds most of the genetic diversity within *H. viridiflavus*, which leads to us recognizing this region as the possible center of the diversification of the species. Similar patterns of intraspecific diversity have been recovered for other Western Mediterranean reptiles, e.g., [18,58,59], reflecting the importance of the Southern European peninsulas on the diversification of Mediterranean taxa [60,61].

Our haplotype network shows distinct geographic distribution patterns of genetic diversity in four major haplogroups, which correspond to the four major genetic units recovered in the phylogenetic tree. The most evident pattern is the star-like shape of the west clade, which shows a total lack of geographic structuring because of one major haplotype (i.e., H1) occurring everywhere across the distribution range of this clade (Figure 2). As recovered in other species (e.g., in some species of *Vipera*, [15]; *Podarcis siculus,* [58]; *Timon lepidus-nevadensis*, [62]), this pattern may reflect a relatively recent bottleneck effect followed by a population expansion. This hypothesis is supported by the demographic analyses conducted in our study, which showed significant results for both neutrality tests (Table 1). A similar star-like shape pattern with a widespread frequent haplotype (i.e., H32) was recovered in the South–East–North subclade within the east clade, with significant neutrality tests also pointing towards a recent population expansion (Table 1). Despite the high frequency of some haplotypes in the other two subclades within the east clade, the neutrality tests are non-significant; thus, there are no signs of recent population expansion. This fact could be related to the small sample size available for these subclades, particularly for the central subclade (*n* = 16). Alternatively, it could reflect a more stable demography along the history of these subclades, a pattern that is suggested for some Mediterranean amphibians (e.g., *Rana graeca*, [63]) and reptiles (e.g., some species of *Podarcis*, [64]).

We used two calibration points that were considered in previous studies to date the diversification times for the main genetic units within *H. viridiflavus* [31]. Consequently, following the divergence between west and east clades at the beginning of the Pleistocene (TMRCA Avg./95% HPD = 2.2/1.789–2.570 Mya), diversification within these two clades occurred at distinct times in the late Pleistocene (Figure 2 and Appendix A): between 0.65 (95% HPD = 0.315–1.022) and 0.09 (95% HPD = 0.011–0.214) Mya for the east clade and the subclades composing it, and at about 0.27 (95% HPD = 0.109–0.45) Mya for the west clade. The Pleistocene is characterized by repeated warming–cooling cycles, which are believed to have affected the intraspecific diversity and structure of many species worldwide, particularly in temperate regions [3,4]. Pleistocene warming–cooling cycles become more similar in intensity and duration at about 0.9–0.8 Mya, which is in the middle Pleistocene transition [65,66]. Intraspecific diversification of many ectothermic taxa in the Mediterranean Basin and surrounding regions have been found to be related to this later period, e.g., [58,67,68]. Although the periods of divergence and diversification found for *H. viridiflavus* mostly predate the past-climatic scenarios used to infer the range dynamics of the species (i.e., the oldest period is the last interglacial, LIG, which occurred about 0.12–0.14 Mya), they are representative of the warming–cooling cycles that occurred in the late Pleistocene and are representative of the posterior warming tendency. Therefore, our paleoclimatic reconstructions are useful for inferring the biogeographic history of the species [18,39].

### 4.2. Biogeographic History

Our ecological models show that the current distribution of *H. viridiflavus* is linked to temperate areas with moderate to high levels of precipitation (Table 2, Appendix A). Despite its widespread occurrence across Western France and mountainous regions such as the Pyrenees, the Alps, the Apennines, and Corsica, the species becomes geographically restricted in its southernmost range (Appendix A), avoiding areas of marked Mediterranean regimes, as is the case in Southern Italy [29]. Considering its thermophilic physiology [27] and the rather temperate character recovered by our ecological models, it is therefore expected that marked changes in the temperature and precipitation conditions, as had occurred in the Pleistocene, would have an impact on the species distribution.

A previous study indicated the role of the cold periods of the Pleistocene as major drivers of the genetic structure and diversity of *H. viridiflavus* [32]. Our model projections of past conditions clearly indicate the potential of the cold and dry conditions of the LGM to restrict the species range to coastal areas in Western Italy and Southern France (Figure 3). However, extremely warm periods such as the LIG would also play an important role in the species range, leading to a pattern of restriction to mountain ranges (e.g., in Italy and Spain) and northern latitudes (e.g., in France; Figure 3). However, they were not considered in previous studies on the species’ evolutionary history, i.e., [32], and Pleistocene interglacial periods have already been indicated as playing a major role in the biogeographic patterns that have been recovered for other temperate taxa, either promoting range expansions (e.g., *Zootoca vivipara*; [69]) or leading to range restrictions (e.g., *V. seoanei*; [15]).

The importance of both the interglacial and glacial periods as drivers of the genetic structure and diversity of the species is clearly recovered in the analysis of climatic stability, which shows very restricted areas of climatic stability for the species when LIG and LGM periods are considered (Figure 4). The stable areas recovered in the LIG–LGM overlap match with the potential climatic refugia for the four genetic groups, as recovered in both the phylogenetic tree and haplotype network (Figure 2), reinforcing a pattern fitting the model of “refugia within refugia” for the Italian peninsula [70]. It is likely, therefore, that the climatic conditions that occurred in the LIG and LGM produced latitudinal/altitudinal range shifts and massive extinctions in the species and that the ancestors of the four recovered genetic units persisted in climatic refugia. The west clade could have persisted in an area south of the Western Alps (between Southeastern France and Northwestern Italy), while the three subclades within the east clade could have persisted in isolated patches of climatic suitability across Sicily and the southern part of the Italian peninsula (Figure 4). Remarkably, similarly located climatic refugia were found for the Western Mediterranean viper *Vipera aspis*, which is distributed across the same area as *H. viridiflavus*, presents similar ecological requirements, and shows a similar genetic structure [18].

Our model projections to the most recent past periods (i.e., the Bølling–Allerød, BA, and the mid-Holocene, MH) indicate a rather similar distribution of climatic suitability for the species to current times (Figure 3). The analysis of climatic stability shows a major increase in stability through the western coast in the overlap region of LGM–BA, which is later followed by a massive increase everywhere inside and outside of the current range of *H. viridiflavus* in the posterior two-time period overlaps (Figure 4). These results would support the likely occurrence of a rapid, recent population expansion for the west clade and South–East–North subclade from climatic refugia, as recovered using genetic data. Therefore, it is likely that the increase in suitable conditions after the LGM allowed for the northwards expansion of the South–East–North subclade from the Southern Italian peninsula and the westwards, northwards, and southwards expansion of the west clade from a climatic refugee located in the South-western Alps.

As can be identified by the distribution of the lineages, topographic barriers such as the Apennines would have likely dictated pathways of post-glacial colonization [71,72], while glacial land bridges such as those that occurred between Sardinia, Corsica, and mainland Italy would allow for the colonization of these Mediterranean islands, e.g., [73]. In addition, other ecological factors not considered in the ecological modelling approach could have played a role in the post-glacial expansion of *H. viridiflavus*. This is the case in interspecific competition with ecologically similar species, which are parapatrically distributed, which was suggested to be the main factor limiting distributions in other European snakes, e.g., [15,18]. In the specific case of *H. viridiflavus*, it is likely that competition with *H. gemonensis* and almost with *Malpolon insignitus* (Geoffroy de Saint-Hilaire, 1827) in the Balkan peninsula and with *Malpolon monspessulanus* (Hermann, 1804) in the Iberian peninsula and Southern France could have prevented further eastward and westward expansions; however, no studies have addressed the potential niche exclusion between these species.

## 5. Conclusions

This study relies on the integration of both phylogeographic and paleoclimatic tools to clarify the role of climate as the driver of the evolutionary history of *H. viridiflavus*. As published in previous studies [31,32,36], there is an evident pattern of a decrease in genetic diversity from east to west across the species range, as almost all of the genetic diversity in *H. viridiflavus* is concentrated in the Italian peninsula. Although this pattern has been previously explained by assuming population retractions and extinctions during glacial periods, we showed that interglacial warm periods also had a significant impact on the species’ distribution and affected its genetic structure and variability. In addition, we provided further insights into the population expansion events that occurred from the southern-located climatic refugia once the climate started to ameliorate after the Pleistocene. We recall that our study is based on a single mtDNA fragment, and future studies should undoubtedly focus on increasing genetic insights (e.g., through multi-locus phylogenomic analyses, e.g., [57]). Overall, our study exemplifies the role of climate in the range dynamics of an ectothermic temperate species, contributing to the understanding of the evolutionary processes that occurred in the Mediterranean Basin hotspot.

## Figures and Tables

**Figure 1 animals-13-02143-f001:**
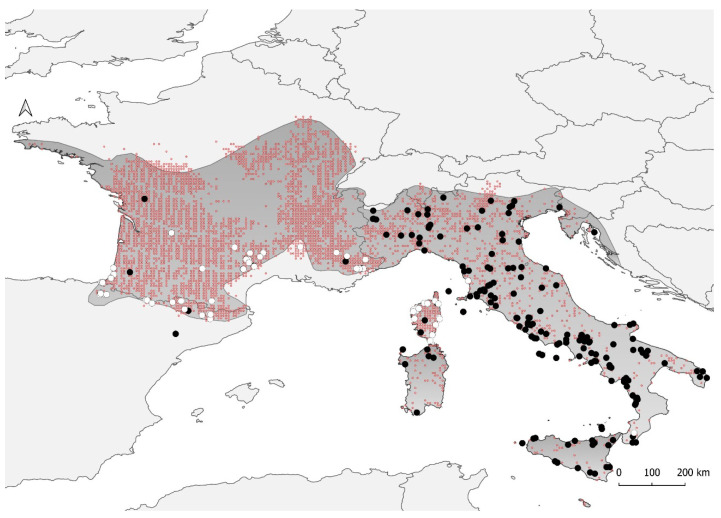
Distribution of the European whip snake, *Hierophis viridiflavus*, according to the IUCN (Vogrin et al., 2009 [37]) and the location of the genetic data used in this study. White dots (*n* = 80) correspond to the samples sequenced in this study, whereas black dots (*n* = 224) refer to the location of the sequences used in previous studies. Small dots in red correspond to the occurrence data used to conduct paleoclimatic models.

**Figure 2 animals-13-02143-f002:**
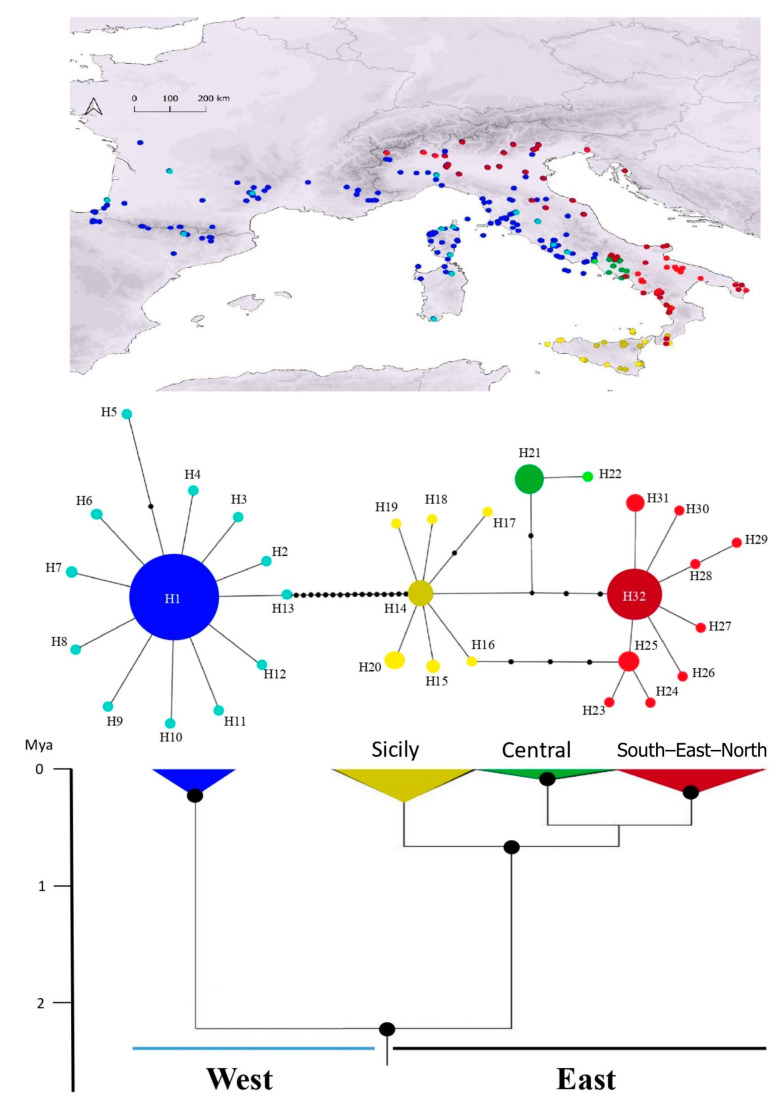
Phylogeographic relationships within *H. viridiflavus* based on mtDNA. The colors in the map (**top**) represent the identified groups recovered in the haplotype network and the dated Bayesian phylogeographic tree (**bottom**). In the haplotype network, each ball represents one mutational step, and haplotypes are signified using distinct letters (see Appendix A for further details). In the phylogeographic tree, black dots represent well-supported nodes (pp > 0.95; see Appendix A for further details).

**Figure 3 animals-13-02143-f003:**
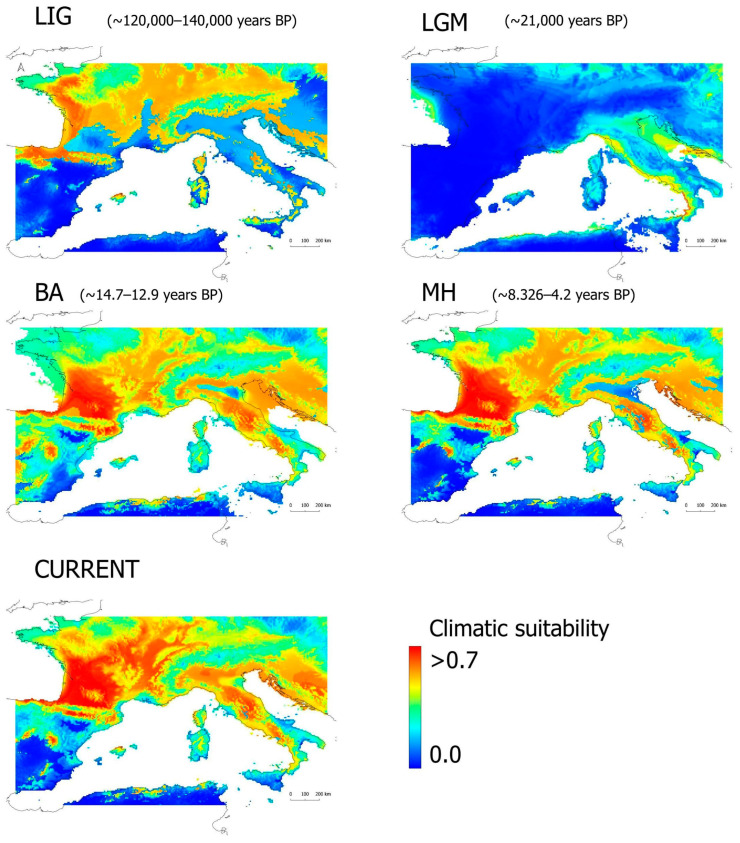
Climatic suitability for *Hierophis viridiflavus* over time. The plots represent the average model predictions for current times (CURRENT) and the projections for four different past periods (LIG, last inter glacial; LGM, last glacial maximum; BA, Bølling–Allerød; MH, mid-Holocene).

**Figure 4 animals-13-02143-f004:**
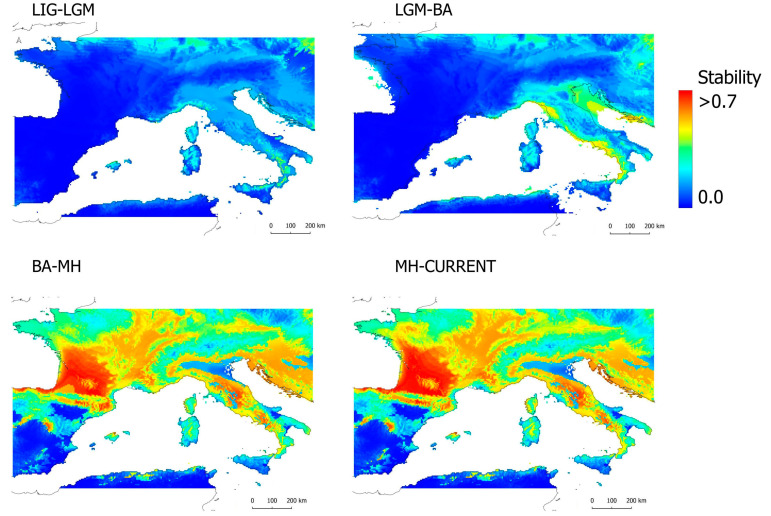
Stability of climatic suitable areas for *Hierophis viridiflavus* over time. In each plot, two consecutive periods are considered: LIG–LGM, last inter glacial–last glacial maximum; LGM–BA, last glacial maximum–bølling–allerød; BA–MH, Bølling–Allerød–mid-Holocene; MH–CURRENT, mid-Holocene–current times).

**Table 1 animals-13-02143-t001:** Number of samples obtained (*n*), number of haplotypes identified (h), haplotype diversity (Hd), nucleotide diversity (π, Tajima’s D and Fu and Li’s D neutrality tests) for the west and east clades and the three eastern subclades. Significant *p*-values are signaled in bold.

Clade–Subclade	*n*	h	Hd	π	Tajima’s D	Fu and Li’s D
West	188	13	0.144	0.00029	**−2.314**	**−5.777**
East	116	19	0.742	0.00491	−0.729	−1.478
South–East–North	76	10	0.464	0.00104	**−1.769**	**−2.967**
Central	16	2	0.125	0.00022	−1.162	−1.453
Sicily	24	7	0.678	0.00162	−1.591	−2.003

**Table 2 animals-13-02143-t002:** Climatic variables used in this work, depicting codes, names, average contribution, and SD of each variable and VIF.

Code	Name	Avg. Contribution ± SD	VIF
BIO3	Isothermality	10.762 ± 1.005	1.184
BIO5	Max Temperature of Warmest Month	34.342 ± 2.515	1.64
BIO12	Annual Precipitation	52.176 ± 2.950	1.502
BIO15	Precipitation Seasonality	2.719 ± 1.864	1.438

## Data Availability

Not applicable.

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
