# Peer review of "Phylogeographic and Paleoclimatic Modelling Tools Improve Our Understanding of the Biogeographic History of Hierophis viridiflavus (Colubridae)"

_animals, 2023, doi:10.3390/ani13132143_

Round 1

Reviewer 1 Report

Dear colleagues, 

I have carefully read your ms and I found it intersting and much promising for the additional info and possible explanation of the biogeographichistory of the European whip snake.

I shortly commented upon the discussion where I found only some minor points that deserve some attention.

In my opinion a suitable ms to be published on Animals.

Best regards,

Marco Zuffi

Reviewer 2 Report

The authors might consider the following comments in making improvements to their manuscript.

0) The manuscript generally reads smoothly, but English grammar, word choice, and style could be improved in several spots in the manuscript.

1) Title, Simple Summary, and Abstract.  Perhaps it would be best to state the common species name for the focus of this study early in the title or abstract so the readers who may not be herpetologists know that the paper is about whip snakes?

2) Introduction.  I generally find attempts to fit intraspecific genetic variation within a species or species complex with niche modeling across time to be a frustrating, somewhat hand-wavy enterprise.  Could the authors be more specific about what interplay there is regarding prediction and corroboration between the two types of data?  Is the goal of the authors to construct a plausible scenario for the evolution of this species and its genetic material in the Pleistocene to the present, or are they testing specific biogeographic or climatic models using the genetic data?  Are they assuming that the niche-modeling information just is, or are there error estimates in the niche modeling (uncertainty) that is dealt with in a credible way.  Overall, I found the Introduction to be a bit vague as to what is being tested and with what data and how, or whether the authors are simply using all available data at their disposal to simply reconstruct, as best as they can, the biogeographic history of this snake species?

3) Climatic Variables.  It might be good to list in this Methods section the 17 bioclimatic variables in the text or in an online table?  Also, where say, "In this work, we extended model projections to an additional period of the Pleistocene (i.e. the Bølling-Allerød period, BA) that enables us to assess the general warming trend that occurred during that period until the Early Holocene [43] that may have affected this species’ historical demography and population structure.

", it might be good to list the age of the Bølling-Allerød period, since the time of all of the other time levels that were examined are already given in this paragraph.

4) line 183.  Almost always, partitioning mt protein-coding genes by codon improves model fit vastly, in particular when rate multipliers are used for branch lengths for each of the three codon positions.  Since the dataset is minimal here, just 80 new sequences from one mt gene, it would be good to test whether partitioning by codon in terms of substitution model, base composition, rate heterogeneity, and branch length improves fit significantly.  Or, the authors could argue why partitioning by codon did not improve fit or why partitioning more finely was not attempted.  Third codons always (always...) evolve way faster than 2nd codons and have much longer branch lengths on a tree, and base composition also generally is significantly different between 2nd and 3rd codons, so it is likely such partitioning would improve fit and inference and divergence time estimates based on the small number of informative sites that were examined in this dataset.

5) line 188.  Secondary calibrations are not ideal, and this approach has been critiqued.  The authors might consider citing some of the literature that has frowned on using secondary calibrations from prior clock studies, and it is also not ideal to just choose arbitrarily priors around point estimates of divergence times rather than trying something potentially better.  For example, the divergence time estimates used from prior clock studies surely have uncertainty that was estimated for each divergence in the prior work.  Why not take this empirically estimated uncertainty into account?  I fear that divergence estimates might be unrealistically narrow unless such uncertainty is taken into account, especially given that no fossil calibrations were used here.

6) line 197.  This network stuff: I see no reason to do it.  Just make an ML or Bayesian tree and collapse nodes with no/extremely low support.  Then, good models can be used to estimate the network.  Haplotypes with zero length branch lengths that descend from a node would then be interpreted as 'ancestral' haplotypes.  Were there large differences between the authors' phylogenetic Bayes trees and their haplotype network trees after collapse of low-support nodes in the Bayesian analysis?  If so, which topology is correct, or are both wrong.  I mean, they cannot both be right, right?  Doing different phylogenetic analyses like this begs the question of which, if any, of the historical inferences is accurate.

7) Figure 2.  How do the authors interpret the connection in their network between H16-yellow and H25 red, which seem to be separated by just three mutations, but connect distantly related sequences in the phylogenetic tree below?  Does this connection have any biological significance (e.g., indicating recombination, sequence convergence, or something else)?

8) line 243.  What is the 95% credibility range for this divergence time.  As noted above, I think if the uncertainty of the calibrations from a prior clock study were taken into account, the uncertainty at this node and others could be much greater, or not?  Is this a problem in the authors' opinion; if not, should explain why?

9) Figure 3.  For comparison to the divergence times in the tree, it might be best to add the times (age before present) for each panel in this figure to help the reader?

10) line 327 paragraph.  It should be noted here that these 'clades' are just clades supported by a single mt gene tree.  For example, if 100 more genes were sequenced, perhaps none would support these clades.  This uncertainty derives from analyzing just one gene in this study; there is no way around this problem until many more nuclear loci are sequenced.  For example, such data might suggest recent gene flow between clades or not even support the mt gene clades shown here?

11) line 350.  I do not do these types of analyses so cannot comment on the meaning of neutrality tests relative to expansion.  I assume here that the authors know what they are doing and that these analyses make sense.

12) line 362.  'Known' not the best word choice as no evolutionary reconstruction is ever really 'known'?  In the best case scenarios, an evolutionary may be very well supported by the data at hand but not really known.

13) line 356 paragraph.  The uncertainty of divergence dates should be noted here in the text somehow.

14) line 381.  Although the authors briefly noted it earlier, it is maybe a little naive to assume that most species do not evolve much over 2 million years of evolutionary time in a changing environment, especially given the extreme shrinkage of decent habitat during recent times in the past.  In such times, it seems there would be strong selection pressure to change ecological niche rather than just shrink down to just a small refugial territory.  I am assuming this is just an accepted problem in the types of analyses done here and in many other studies?

15) line 397.  The authors say, "The stable areas recovered in the LIG-LGM overlap match with potential climatic refugia for the four genetic groups as recovered in both the phylogenetic tree and haplotype network, reinforcing a pattern fitting the model of “refugia within refugia” for the Italian Peninsula [70] ".  I mean, if I squint at the colored maps I can maybe sort of see something like this. ..or not?  I wonder whether the authors' interpretation is partly driven by wishful thinking in looking for patterns that may or may not correspond between the molecular genealogical pattern and the niche modeling pattern?  There is really no explicit statistical method given for why the authors strongly think there is congruence between the patterns.  Maybe the authors could overlap the locations of the genetic clades with the niche modeling maps to better show this correspondence to clades in the 'hot spots' for this species in the niche modeling maps?

The word usage and style could be improved a bit, but the paper reads pretty well.  There are just some awkward phrases in places and a few grammatical errors.
